# Sympatric Populations of the *Anopheles gambiae* Complex in Southwest Burkina Faso Evolve Multiple Diverse Resistance Mechanisms in Response to Intense Selection Pressure with Pyrethroids

**DOI:** 10.3390/insects13030247

**Published:** 2022-02-28

**Authors:** Jessica Williams, Victoria A. Ingham, Marion Morris, Kobié Hyacinthe Toé, Aristide S. Hien, John C. Morgan, Roch K. Dabiré, Wamdagogo Moussa Guelbéogo, N’Falé Sagnon, Hilary Ranson

**Affiliations:** 1Vector Biology Department, Liverpool School of Tropical Medicine, Pembroke Place, Liverpool L3 5QA, UK; jessica.williams@lstmed.ac.uk (J.W.); victoria.ingham@uni-heidelberg.de (V.A.I.); marion.morris@lstmed.ac.uk (M.M.); john.morgan@lstmed.ac.uk (J.C.M.); 2Centre National de Recherche et de Formation sur le Paludisme (CNRFP), Ouagadougou 01 BP 85, Burkina Faso; toe_kob@yahoo.fr (K.H.T.); guelbeogo.cnrfp@fasonet.bf (W.M.G.); n.fale.cnlp@fasonet.bf (N.S.); 3Laboratory of Fundamental and Applied Entomology, Université Joseph KI-ZERBO, Ouagadougou 01 BP 85, Burkina Faso; 4Institut de Recherche en Sciences de la Santé (IRSS), Bobo-Dioulasso 01 BP 390, Burkina Faso; aristide.hien@yahoo.fr (A.S.H.); dabireroch@gmail.com (R.K.D.)

**Keywords:** malaria vector, insecticide resistance, insecticide treated nets, cytochrome P450s, kdr, cuticular resistance

## Abstract

**Simple Summary:**

Targeting mosquitoes with insecticides is one of the most effective methods to prevent malaria transmission. Although numbers of malaria cases have declined substantially this century, this pattern is not universal and Burkina Faso has one of the highest burdens of malaria; it is also a hotspot for the evolution of insecticide resistance in malaria vectors. We have established laboratory colonies from multiple species within the An. gambiae complex, the most efficient group of malaria vectors in the world, from larval collections in southwest Burkina Faso. Using bioassays with different insecticides widely used to control public health pests, we provide a profile of insecticide resistance in each of these colonies and, using molecular tools, reveal the genetic changes underpinning this resistance. We show that, whilst many resistance mechanisms are shared between species, there are some important differences which may affect resistance to current and future insecticide classes. The complexity, and diversity of resistance mechanisms highlights the importance of screening any potential new insecticide intended for use in malaria control against a wide range of populations. These stable laboratory colonies provide a valuable resource for insecticide discovery, and for further studies on the evolution and dispersal of insecticide resistance within and between species.

**Abstract:**

Pyrethroid resistance in the Anopheles vectors of malaria is driving an urgent search for new insecticides that can be used in proven vector control tools such as insecticide treated nets (ITNs). Screening for potential new insecticides requires access to stable colonies of the predominant vector species that contain the major pyrethroid resistance mechanisms circulating in wild populations. Southwest Burkina Faso is an apparent hotspot for the emergence of pyrethroid resistance in species of the Anopheles gambiae complex. We established stable colonies from larval collections across this region and characterised the resistance phenotype and underpinning genetic mechanisms. Three additional colonies were successfully established (1 An. coluzzii, 1 An. gambiae and 1 An. arabiensis) to add to the 2 An. coluzzii colonies already established from this region; all 5 strains are highly resistant to pyrethroids. Synergism assays found that piperonyl butoxide (PBO) exposure was unable to fully restore susceptibility although exposure to a commercial ITN containing PBO resulted in 100% mortality. All colonies contained resistant alleles of the voltage gated sodium channel but with differing proportions of alternative resistant haplotypes. RNAseq data confirmed the role of P450s, with CYP6P3 and CYP6Z2 elevated in all 5 strains, and identified many other resistance mechanisms, some found across strains, others unique to a particular species. These strains represent an important resource for insecticide discovery and provide further insights into the complex genetic changes driving pyrethroid resistance.

## 1. Introduction

Pyrethroid insecticides have played a key role in interrupting malaria transmission. All insecticide treated nets (ITNs) in use contain pyrethroids; they are the major active ingredient in insecticidal household aerosol sprays and coils and, prior to the advent of widespread resistance, they were the preferred chemistry for use in indoor residual spraying programmes [1]. Malaria vectors will also likely encounter pyrethroids in their aquatic habitats as this insecticide class is still widely used in agriculture, and mosquito breeding sites in rural areas frequently contain detectable levels of insecticides utilised to spray nearby crops [2,3].

Resistance to pyrethroids was first detected in African malaria vectors in the 1970s [4] and is now widespread [5], prompting the search for new chemistries for use in vector control tools. Whether re-purposing chemistries used to control other pest species, or searching for new insecticide classes, the identification of suitable chemistries requires a robust screening pipeline that includes screening potential compounds against a range of mosquito populations resistant to current chemistries [6,7]. Whilst ultimately testing against natural wild populations will be required, the availability of stable laboratory colonies of the predominant vector species, containing the major resistance mechanisms circulating in the field can greatly accelerate the insecticide screening pipeline by identifying resistance liabilities at an early stage [8].

We have previously described the properties of several colonies of *Anopheles* mosquitoes that have been widely used in insecticide discovery programmes; these contain well characterised target site mutations and metabolic resistance conferred by elevated levels of specific pyrethroid metabolising cytochrome P450s [9]. However, recent studies on *Anopheles gambiae* s.l. mosquito populations from West Africa have identified additional, potent pyrethroid resistance mechanisms such as reduced penetration caused by cuticular thickening [10,11], insecticide sequestration by pyrethroid binding proteins in the mosquito appendages and novel resistance associated haplotypes of the pyrethroid target site, the voltage gated sodium channel (VGSC) [12,13,14]. Several of these resistance mechanisms could potentially cause cross resistance to existing or new classes of insecticides; thus, we sought to establish new colonies of pyrethroid resistant *An. gambiae* s.l. from Burkina Faso, stabilise and quantify their pyrethroid resistance phenotypes and determine the underpinning mechanisms responsible for resistance.

*An. gambiae* is a species complex of at least nine morphologically identical species. Three of these (*An. gambiae* s.s, *Anopeheles coluzzii* and *Anopheles arabiensis*) are amongst the most important malaria vectors and are found in Burkina Faso [15]. Introgression of genes under selection pressure is not uncommon between members of the complex with several well documented cases of exchange of haplotypes containing point mutations in insecticide target sites [16,17]. The Southwest region of Burkina Faso is an important agricultural region of the country and also an area of stubbornly persistent malaria transmission, perhaps partially linked to the exceptionally high levels of pyrethroid resistance in the malaria vectors from this region [18,19]. We established three new colonies from larval collections in the Cascades and Southwest regions of Burkina Faso between 2015 and 2018, encompassing each of the three members of the *An. gambiae* complex found in the country. Phenotyping and molecular characterisation of these new colonies, the previously established Banfora M colony (Cascades region) and the VK72014 colony (neighbouring Hauts Basin region), revealed high levels of pyrethroid resistance with four colonies meeting the WHO definition of high intensity resistance and the fifth with moderate intensity. Genotyping and RNAseq identified resistance mechanisms in common between strains but also key differences that may have implications for susceptibility to alternative insecticide classes.

## 2. Materials and Methods

### 2.1. Establishment of Strains

Details of the strains used in this study are provided in Table 1. The origins of the susceptible strains Kisumu and Moz and the pyrethroid resistant Burkina Faso populations VK7 2014 and Banfora M have been described previously [9]. Larval collections from multiple villages in the Comoé Province, Cascades region of Burkina Faso in 2015 led to the establishment of two strains: Bakaridjan and Banfora. Briefly, larvae were reared to adults, allowed to mate and then females transferred to Eppendorf tubes to oviposit individually as described previously [20]. Females were killed by freezing after oviposition. Dried females, and egg papers were transported to the Liverpool School of Tropical Medicine. Species ID on the F0 female was performed [21] and egg batches from *An.*
*gambiae* (s.s.) or *An. coluzzii* females were pooled to establish two separate colonies. The *An. coluzzii* colony was named ‘Banfora’ after the Banfora district as the colony was established from collections from several villages within this district (Tiefora, Pont Maurice, Sikane and Djomale; Figure 1). The *An. gambiae* s.s. strain was named ‘Bakaridjan’ as the majority of egg batches used to establish this strain were collected from this village. The *An. coluzzii* Tiefora strain and the *An. arabiensis* Gaoua-ara strains were established as above from larval collections performed in Tiefora Village Comoé Province, Banfora District and Gaoua District, Poni Province in 2018. The insecticide-susceptible colony N’Gousso originated from Cameroon [22].

### 2.2. Mosquito Rearing

Insectaries were maintained under standard conditions at 26 ± 2 °C and 70% relative humidity ±10% under L12:D12 h light:dark photoperiod. All stages of larvae were fed on ground fish food (TetraMin^®^ tropical flakes, Tetra^®^, Blacksburg, VA, USA) and adults were provided with 10% sucrose solution ad libitum.

### 2.3. Selection and Resistance Profiling

The five insecticide resistant strains were routinely selected every 3rd to 5th generation with 0.05% deltamethrin to preserve their resistant phenotype. Insecticide papers were purchased from the WHO facility at the University Sains Malaysia (USM), Penang, Malaysia and used a maximum of 6 times following the WHO procedure [24]. Selection was undertaken at the adult stage (2–5 days old) using the WHO susceptibility bioassay [24]. Exposure times varied between strains to ensure at least 50% survival (VK7 2014 2 h, Banfora M and Bakaridjan 2–3 h, Gaoua-ara 2–4 h and Tiefora 4–5 h). All adults from the generation to be selected were exposed, with results scored from at least 100 individuals. Following exposure, the mosquitoes were transferred to holding tubes and supplied with 10% sucrose solution and the initial knockdown effect was scored immediately post exposure. At 24 h post exposure, mortality rates were recorded. Bioassays and 24 h holding periods were conducted at 26 ± 2 °C and 80 ± 10% RH.

Each strain was profiled annually against eight insecticides (except VK7 2014 which was profiled against six insecticides) representing the different insecticide classes currently used for mosquito control, to monitor the stability of their resistance phenotype; as described in [9] insecticides used were permethrin, deltamethrin, alpha-cypermethrin, DDT, dieldrin, bendiocarb, propoxur and fenitrothion. Results for VK7 2014 and Banfora M have been reported previously [9], but are included here for comparative purposes.

The intensity of resistance was evaluated in the different strains using papers treated with 5× and 10× the diagnostic dose of permethrin following the WHO procedure [24].

### 2.4. Synergist Bioassays

The impact of the synergist piperonyl butoxide (PBO) on pyrethroid induced mortality in each of the resistant strains was assessed in two separate experiments. Firstly, 2–5 day old female mosquitoes were pre-exposed to PBO papers impregnated with PBO (4%) followed by 1, 2, 3 or 4 h exposures to papers impregnated with permethrin (0.75%) according to the WHO protocol [24].

In the second experiment, mortality rates following sequential PBO then pyrethroid exposure were compared with simultaneous exposure to insecticide and synergist. Adult females from three strains were exposed to either (1) a pyrethroid only 1-h exposure; (2) a 1-h PBO pre-exposure followed by a 1-h pyrethroid exposure, or (3) a 1-h combination exposure (with PBO and either pyrethroid on the same paper). These experiments were performed separately using 0.75% permethrin papers and 0.05% deltamethrin papers.

In both experiments, solvent only paper (no AI) and a PBO control, where a 1-h PBO exposure was followed by 1-h blank exposure were included. Differences in mortality with and without PBO exposure were analysed for significance using Fisher’s exact test.

### 2.5. Cone Bioassays

Mosquitoes were exposed to PermaNet^®^ 3.0 LN (Vestergaard Frandsen SA, Denmark) a LLIN consisting of a top panel made of monofilament polyethylene (100 denier) fabric incorporating deltamethrin at 4 g/kg (approx. 180 mg/m^2^) and piperonyl butoxide at 25 g/kg (approx. 1.1 g/m^2^), plus side panels made of multifilament polyester (75 denier) fabric with a strengthened border treated with deltamethrin at 2.8 g/kg (approx. 118 mg/m^2^) in WHO cone bioassays [25]. Following net airing of 2 weeks, pieces of netting (25 cm × 25 cm) were cut from the roof and side of the PermaNet 3.0 and cohorts of approximately 50 mosquitoes of each strain were exposed using the WHO standard protocol. Controls were exposed to insecticide free net in two replicates, each with 5 mosquitoes, one just before and one just after the treated exposures. Following exposure, the mosquitoes were aspirated into paper cups and supplied with 10% sucrose solution, and the initial knockdown effect was scored at 1 h and mortality was scored at 24 h post exposure.

### 2.6. Target Site Mutation Genotyping

Genomic DNA was collected within the first 5 months of colonisation of each strain and every subsequent 6–12 months thereafter. The DNA was extracted from 48 non-blood-fed females using a Qiagen blood and tissue DNA extraction kit (Qiagen, Germantown, MD, USA). Species ID was identified using the SINE PCR protocol [21].

Each strain was genotyped to identify the frequency of known target site resistance alleles (alleles 995F, 995S and 1570Y in the VGSC, the *ace-1*119S allele and the RDL alleles 296G and 296S) using Taqman™ assays [26,27,28,29]. The allelic variant 114T of the glutathione transferase *GSTe2* gene was also genotyped as previously [30].

### 2.7. RNAseq Transcriptomic Analysis

RNA was extracted from pools of 5, 3–5 day old presumed-mated adult females, snap frozen in the −80 °C at 10 am, using a PicoPure kit (Applied Biosystems Thermo Fisher, Waltham, MA, USA, after homogenisation with a motorised pestle. Quality and quantity of the RNA was analysed using an Agilent TapeStation (Santa Clara, CA, USA) and Nanodrop (Thermo Fisher) respectively, and three (Moz, Gaoua-ara, N’Gousso, Tiefora) or four (Banfora, VK72014, Kisumu, Bakaridjan) replicates from each strain sent for sequencing at Centre for Genomics, Liverpool, UK (RNA extractions for Banfora were performed as part of a separate study [31] but using the same methodology).

The resulting data was run through appropriate QC using FastQC and aligned to the latest *Anopheles gambiae* s.l. genome assembly PEST4 using Hisat2 with default parameters. The resulting bam file was sorted using samtools and the number of reads aligned to each gene extracted using featureCounts. Over 70% read assignment was seen for each replicate of each population with the majority showing >85%. Data from the *An. gambiae* s.s and *An. coluzzii* resistant populations were compared to the two susceptible populations (Kisumu and N’Gousso) using limma. First, a model matrix was defined to account for the populations and then contrasts were made to compare the resistant *An. gambiae* and *An. coluzzii* to both susceptible populations through the function makeContrasts using resistant—(N’Gousso + Kisumu)/2. Counts were then transformed to log2 counts per million reads (CPM), residuals calculated, and a smoothed curve fitted using the voom function which utilises normalisation factors calculated using calcNormFactors. lmFit was used to fit a linear model for each gene and eBayes used to smooth the standard errors. The function topTable was then used to retrieve results and written out to file; significance was taken as adjusted *p* value ≤ 0.05. In the case of the single *An. arabiensis* population, the contrast matrix was simply a resistant vs. susceptible design. In each instance the filterByExpr function from the EdgeR package was used to remove genes with low read number. Enrichment analysis was performed using the built-in GO term enrichment analysis on VectorBase with a Benjamini significance cut-off of ≤ 0.05. Revigo was then used to remove redundant GO terms allowing more appropriate visualisations; default parameters were used with a 0.5 selection. A custom table was also used with hypergeometric tests with fdr cut-off of ≤0.05 to integrate KEGG, Reactome and a priori genes of interest into the enrichment analysis (https://github.com/VictoriaIngham/BurkinsStrains) (accessed on 9 December 2021). All RNAseq data is deposited in SRA under accession PRJNA780362 and PRJNA750256.

### 2.8. Metabolic Resistance—Detox Gene Expression Levels

One to four μg of RNA, extracted from three pools of 5, 3–5-day-old female as described above, was reverse transcribed using Oligo dT (Invitrogen, Warrington, UK) and Superscript III (Invitrogen). The resulting cDNA was diluted to 4 ng/µL and used as a template in the subsequent PCR reactions. Primers and probes as described by Maviridis et al. [32] were ordered from Integrated DNA Technologies (Leuven, Belgium), with Cy5 replacing Atto647N. Primers and probes were diluted to 10 µM for use in a 10 µL final reaction. Four multiplex reactions were carried out on each cDNA set in technical triplicate, as follows: (i) CYP6P4, CYP6Z1 and RPS7; (ii) CYP4G16 and CYP9K1; (iii) CYP6M2 and CYP6P1; (iv) CYP6P3 and GSTE2. PrimeTime Gene Expression Master Mix (Integrated DNA Technologies) was used to set up each reaction following the manufacturer’s instructions. Each reaction was carried out on a MxPro 3005P qPCR System (Agilent) with the following thermocycling conditions: 3 min at 95 °C followed by 40 cycles of 15 s at 95 °C; 1 min at 60 °C. Cycle threshold (Cq) values were exported and analysed using the ΔΔct methodology [33], using RPS7 as an endogenous control. Gaoua-ara were normalised against the susceptible Moz strain of *An. arabiensis*, and Bakaridjan, Banfora M, Tiefora and VK7 2014 were compared to both N’Gousso and Kisumu. A homogeneity of variance test was used to determine if data were normally distributed. Δct values were transformed to normalise (where applicable) and an ANOVA test, followed by Dunnett’s test was performed. Where transformations did not normalise the data, a Dunn test was performed.

## 3. Results

### 3.1. Discriminating Dose Assays

Bakaridjan, Gaoua-ara, Banfora M, Tiefora and VK7 2014 are all resistant to pyrethroids and DDT according to WHO definitions [24] (Figure 2). Gaoua-ara and Tiefora are also resistant to the organochlorine dieldrin. Bakaridjan, Gaoua-ara and Tiefora are resistant to the carbamates propoxur and bendiocarb with Banfora M resistant to bendiocarb only. None of the five strains are resistant to the organophosphate fenitrothion. Kisumu and Moz are susceptible to all the insecticides tested and results have been reported previously [9]. N’Gousso showed less than 90 % mortality after exposure to propoxur (87%), DDT (61%) and dieldrin (39%) but was susceptible to other insecticides tested.

The results of profiling the five resistant strains against 5 and 10× diagnostic dose (DD) of permethrin are shown in Figure 3. All 5 strains survived exposure to 5× DD (mortality ranged from 14% to 71%). Four of the strains also showed less than 90% mortality after exposure to 10 × permethrin papers (and would be described by WHO as having high intensity resistance) whereas Gaoua-ara with 55% mortality with 5× papers, 98% mortality with 10× is defined by WHO as moderate to high intensity resistance.

### 3.2. Impact of PBO on Pyrethroid Mortality

All strains showed significant synergism when pre-exposed to PBO followed by a 4-h exposure to permethrin but synergism was not consistently observed with shorter pyrethroid exposures and PBO pre-exposure did not fully restore susceptibility to permethrin in any of the strains (Figure 4; full mortality results and synergism ratios are available in Appendix A). The highest synergist ratios were seen for Banfora where significant synergism was observed at all permethrin exposures greater than 2 h and PBO:permethrin synergism ratios ranged from 7:1 (2 h) to 54:1 (3 h). Negative controls (both control papers only and PBO followed by control papers) gave <4% mortality in all assays.

In a separate set of experiments the effect of sequential versus simultaneous exposure to PBO and pyrethroids was compared (Appendix A) with pyrethroid exposure duration constant at one hour. PBO did not synergise permethrin in these experiments but the efficacy of deltamethrin was significantly improved in all three strains with both PBO exposure methods (*p* < 0.0001 in all cases). Simultaneous exposure to PBO and pyrethroids results in increased mortalities compared to PBO pre-exposure for all three strains but this was only significant (*p* < 0.05) in Bakaridjan for both insecticides and in Banfora with deltamethrin. Full mortality results and synergism ratios are available in additional Appendix A.

### 3.3. Cone Bioassays

Exposure to the side of the PermaNet 3.0 net in a cone bioassay consistently resulted in <10% mortality for all 5 strains but exposure to the roof (containing PBO) resulted in >98% mortality (Figure 5).

### 3.4. Target Site Mutation Genotyping

All the strains were screened for five target site mutations and one mutation in a detoxification gene (Figure 6). The 995F kdr allele was fixed in Bakaridjan and VK72014, but was present at quite low frequencies in Tiefora (allele frequency 0.06) and Banfora M (allele frequency 0.38). The *An. arabiensis* Gaoua-ara strain contained both 995F and 995S with allele frequencies of 0.49 and 0.45, respectively. The 1570Y kdr allele was detected in Bakaridjan, VK7 2014, Banfora M and Tiefora with allele frequencies of 0.04, 0.35, 0.48 and 0.04, respectively. The ace-1 mutation was absent from all strains except a very low frequency in the Tiefora strain. The RDL 296S allele was detected in Gaoua-ara, Banfora M, VK7 2014, and Tiefora with allele frequencies of 0.65, 0.22, 0.03 and 0.26, respectively; only the wildtype form of A296 was found in the *An. gambiae* Bakaridjan strain. The GSTE2 114T detox gene modification was found in Banfora M, VK7 2014 and Tiefora with allele frequencies of 0.66, 0.77 and 0.46, respectively.

### 3.5. RNAseq Analysis

RNAseq analysis was carried out on a minimum of three biological replicates from the five resistant strains and the three laboratory susceptible colonies, Kisumu, N’Goussu and Moz. The correlation matrix shows high degrees of similarity between the two *An. arabiensis* populations, Gaoua-ara and Moz but no clear segregation according to species for the *An. gambiae* and *An. coluzzii* strains (Appendix A). Hence, for all further analysis of differential expression between resistant and susceptible strains, Gaoua-ara was compared to Moz alone whereas the three *An. coluzzii* (Tiefora, VK72014, Banfora) and one *An. gambiae* s.s.(Bakaridjan) resistant strains were compared to the average values from the two susceptible strains of *An. coluzzii* (N’Gousso) and *An. gambiae* (Kisumu).

### 3.6. Similarities between Strains

The total number of genes differentially expressed across all the resistant compared to susceptible strains is shown in Appendix A. A total of 81 transcripts were up regulated in resistant versus susceptible strains with 73 down regulated. The upregulated transcripts show no enrichment but two P450s known to bind and/or metabolise pyrethroids (CYP6P3 and CYP6Z2) are amongst the most highly upregulated genes and two glucoronosyl transfearses (UGT302H2 and UGT306A2) are also elevated in all strains. Down regulated transcripts are strongly enriched for RNA processing (*p* = 1.25 × 10^−4^) and do not contain genes previously associated with pyrethroid resistance.

GO term enrichment was explored for each individual resistant population. Whilst no GO terms were enriched across all five resistant populations, a number of similarities were seen across the four resistant *An. gambiae* and *An. coluzzii* colonies (Appendix A). GO terms significant in up-regulated genes across each population include oxidoreductase activity, typically seen in resistant colonies [34,35] and related to cytochrome p450 activity, and terms related to neuronal signalling, potentially indicating changes in signalling and neurotransmitter activity are associated with resistance to these neurotoxic insecticides. Additionally, terms related to ATPase activity and GPCR signalling, both previously linked to insecticide resistance [36,37] are seen. There are similarities in GO enrichments in the down-regulated subset of genes, with terms related to transcription factor activity, translational regulation, regulation of dephosphorylation and phosphatase complexes, all repressed (Appendix A).

### 3.7. Differences between Strains

The RNAseq data was then interrogated to identify both pathways and *a priori* candidate genes enriched in the up or down regulated genes in each resistant strain with *An. gambiae* and *An. coluzzii* compared to two susceptible controls. Analysis at the individual gene level revealed key differences between the strains. For example, 23 P450s are differentially expressed in one or more strains; as mentioned above, CYP6P3 and CYP6Z2 are up-regulated in all resistant strains but other known pyrethroid metabolisers including CYP6M2, CYP6P2, P4 and P5 and CYP9J5 and 9K1 [38,39] are also up-regulated in two or more strains (Figure 7). This analysis also identifies a number of additional P450s that are highly up-regulated in multiple strains but have not yet been functionally characterised (e.g., CYP4H genes) which merit further study.

Recently, a number of genes with putative roles in sequestering pyrethroids were found to be over expressed in pyrethroid resistant populations from West Africa [40]. RNAseq data from the *An. gambiae* complex in Burkina Faso is supportive of a putative role of hexamerins in pyrethroid resistance in *An. arabiensis* and the VK7 strain of *An. coluzzii* (as shown previously [37]) (Figure 8). Suppression of the hexamerin AGAP001659 (highly upregulated in Gaoua-ara in this study) was previously associated with a reduction in pyrethroid resistance [37]. In addition, several alpha- cyrstallins were up-regulated in one or more of the pyrethroid resistance populations, with this gene family particularly enriched in the Banfora strain of *An. coluzzii* in agreement with earlier qPCR data [40]. Suppression of the alpha-cyrstallin AGAP007159, which is upregulated in multiple Burkina populations, has also been shown to result in a reduction in the resistance phenotype in VK7 2014.

Finally, we looked at expression of genes recently implicated in the cuticular hydrocarbon (CHC) synthetic pathway. This gene list was derived from transcripts encoding the six gene families (propionyl co A synthases, fatty acid synthases, elongases, desaturases, reductases and P450 decarbonylases) that are enriched in the sub epidermal oenocyte cells responsible for CHC production [41]. Several genes in this pathway were up-regulated in the Banfora, Bakaridjan and Gaoua-ara strains but, surprisingly, down-regulated in two of the strains, Tiefora and VK7 2014 (Appendix A). To date only two genes in this putative pathway have been functionally validated, CYP4G16 [42] and the fatty acid synthase FAS1899 [41]; both of these genes are upregulated in the pyrethroid resistant *An. arabiensis* strain (fold changes of 5.2 and 2- fold respectively) suggesting that cuticular resistance may be a particularly important resistance phenotype in this population.

### 3.8. Evaluation of a Multiplex Gene Expression Assay for Metabolic Resistance

RNAseq analysis provided a list of putative genes and pathways potentially contributing to the pyrethroid resistance phenotype in the different strains. However, simpler robust assays of gene expression are needed to further investigate the association between gene expression and resistance phenotype. To this end, the Taqman multiplex assay [32] was used to quantify relative expression of a subset of 8 insecticide detoxification genes in each of the resistant strains compared to their susceptible counterparts (to facilitate correlations with RNAseq data, expression levels from the *An. gambiae* and *An. coluzzii* resistant strains were compared to the average expression of the equivalent transcripts in the *An. gambiae* and *An.*
*coluzzii* susceptible strains). The data generated in this study agreed well with previous Taqman multiplex P450 expression data for VK7, with the exception of CYP9K1 (where significant up-regulation was not detected in earlier generations). P450 levels in Banfora appear more variable between generations, consistent with recent findings that the resistance phenotype is less stable in this population than in other laboratory colonies [28]. Within the current study, there is generally good agreement between the qPCR (Appendix A and Appendix A) and RNAseq data, with the exception of CYP6P3 and CYP6Z1 (Table 2).

## 4. Discussion

This study provides a detailed description of the extent and causes of pyrethroid resistance in three new colonies of *An. gambiae* s.l. from Burkina Faso and provides further information on the genetic basis of pyrethroid resistance in two colonies originating from the same region and described previously [9].

The high levels of pyrethroid resistance present in all five resistant strains, from three different species, reinforces the view that Burkina Faso is a hotspot of resistance [3,43,44,45]. All colonies were maintained under deltamethrin selection and data from WHO intensity assays show little difference in resistance levels between the strains. Although technically the *An. arabiensis* colony is defined as moderately resistant whereas the four *An. coluzzii* and *An. gambiae* strains meet the definition of high resistance, when time of exposure, rather than concentration of insecticide, was the variable, the *An. gambiae* s.s strain was the least resistant of the strains. Bioassays conducted in Burkina Faso in 2010 found that both *An. gambiae* and *An. coluzzii* were significantly more likely to survive permethrin exposure than *An. arabiensis* [46]; these species differences now seem to have been largely eroded, at least in the Burkina Faso populations assayed in this study. Several of the strains also showed resistance to other insecticide classes including carbamates and the cyclodiene, dieldrin. These insecticides are not used for mosquito control in this region and hence the observed resistance may be indicative of agricultural exposure selecting for resistance [3] (or alternatively may be explained by cross resistance between insecticide classes, see below). Insecticides from additional classes including the neonicotinoids and pyrrole, are now being incorporated into vector control products such as indoor residual sprays and ITNs and work is ongoing to assess the susceptibility of these laboratory colonies to these active ingredients. Encouragingly, all strains appear susceptible to the pyrrole chlorfenapyr, used in the ITN IG2^®^ (BASF, Germany) that is being deployed in pilot schemes in Burkina Faso [47].

Pre-exposure to the synergist PBO, did increase permethrin induced mortality but could not fully restore susceptibility in any strain. Simultaneous exposure to PBO and pyrethroids typically resulted in higher mortalities than observed after sequential exposure, perhaps indicating that PBO acts as an adjuvant, as well as an inhibitor of P450s, as has been proposed previously [48] but mortality rates were still well below 100% mortality. However, when mosquitoes from all five strains were exposed to a formulated product containing PBO (the roof of a Permanent 3.0 ITN) 100% mortality was observed after just a 3 min exposure. This highlights the challenges of interpreting results from different bioassays and extrapolating to field effectiveness. High mortalities after exposure to ITNs containing PBO has been observed previously in cone bioassays on *An. coluzzii* from this region and experimental hut studies conducted the same year (2014) showed that PBO ITNs caused higher mosquito mortalities than standard pyrethroid only ITNs [49]. However, rising levels of pyrethroid resistance in the region, appear to be undermining the effectiveness of PBO nets (WMG, N’FS, unpublished data).

As expected, mutations in the VGSC gene, the target site of pyrethroids, were found in all strains, but there was a surprising variation in the frequency of the ‘typical’ kdr haplotypes, 995F and 995S. The 995S allele was only found in *An. arabiensis* and was found in approximately equal frequency to the 995F allele, with the most prevalent genotype being 995F/995S heterozygotes. Similar heterozygotes have been detected in Cameroon and Gabon, with some evidence of a fitness advantage [13]. The 995S allele was first reported in *An. arabiensis* in Burkina Faso in 2008 [46] and the reasons it remains confined to this member of the complex are unknown. The *An. gambiae* Bakaridjan strain and *An.*
*coluzzii* VK7 2014 are both fixed for the 995F allele but this SNP was found at very low frequencies in the other two resistant *An. coluzzii* strains. Subsequent further investigations have detected an alternative VGSC haplotype in pyrethroid resistant *An. coluzzii* from Burkina Faso, consisting of a double mutation at codons 402 and 1527 [14] and have shown that the Banfora M and Tiefora laboratory colonies contain high frequencies of this 402L:1527T haplotype, which is mutually exclusive with the 995F haplotype. The functional significance of the two alternative VGSC resistance haplotypes is the subject of ongoing investigations, comparing the resistance phenotype and fitness costs, and genotyping resistant mosquitoes from neighbouring regions, to try and establish why there is an apparent evolutionary shift away from 995F to alternative amino acid substitutions in these *An coluzzii* populations. In the context of the current study, it is interesting that the 402L:1527T haplotype is only present in one species of the colonies of *An. gambiae* s.l. that were established from the same larval collections in the same breeding sites (Bakaridjan and Banfora M). Introgression of kdr alleles between members of the *An. gambiae* complex has occurred on multiple occasions [50] and longitudinal monitoring of the frequency of these alternative haplotypes in the Cascades region of Burkina Faso may provide an opportunity to monitor any further genetic exchange in this genomic region.

The three new strains described in the current study all contain some level of carbamate resistance, but the target site allele Ace-1 is absent in two of the strains and found at very low frequencies in the third (Tiefora). The persistence of carbamate resistance in these strains for multiple generations in the insectary, in the absence of carbamate selection, together with the absence of target site resistance, point to possible cross resistance between pyrethroids and carbamates. CYP6P3, which is elevated in all of the resistant strains, has been shown to metabolise a wide range of insecticides from different classes, including the carbamate bendiocarb [39,51].

The ‘resistance to dieldrin’ Rdl allele 296S is found at frequencies exceeding 20% in the three newly described strains and its frequency broadly correlates with the prevalence of dieldrin resistance in these strains, with Gaoua-ara (Rdl frequency 0.65) the most resistant to dieldrin. The point mutation GSTE2–114T, which results in an enhanced version of the detox gene Gste2 known to metabolise DDT [30], was found in the three *An. coluzzii* strains at relatively high frequencies (above 0.46 in all cases). All of these strains are highly resistant to DDT; however, the contribution of the GSTE2-114T allele to DDT resistance is difficult to assess in these strains given the presence of target site resistance and the finding that expression levels of GSTE2 are elevated in these resistant strains

RNAseq was used to identify additional resistance mechanisms potentially contributing to the intense pyrethroid resistance phenotype in these strains. The up-regulation of several P450s, together with the partial synergism conferred by PBO, confirmed the importance of this mechanism with many of the known pyrethroid metabolisers up-regulated in multiple strains and the three subfamilies (6P, 6M and 6Z) most widely associated with pyrethroid resistance amongst [39] the most up-regulated, particularly in the *An. coluzzii* strains. Interestingly, in the *An. arabiensis* and *An. gambiae* populations, some of the strongest candidates, based on expression levels alone, are found in other subfamilies of P450s, notably the CYP4H family for *An. arabiensis* which has been implicated in pyrethroid resistance in previous microarray studies [23,52,53] but has not, to our knowledge, been functionally characterised.

In addition, genes thought to play a part in the synthesis and deposition of hydrocarbons on the mosquito cuticle [41] were up-regulated in some strains. Elevated levels of cuticular hydrocarbons have been associated with pyrethroid resistance in *An. coluzzii* mosquitoes from Valle du Kou [10,54] in Burkina Faso and evidence of an association between epicuticle thickness and insecticide resistance has been reported in several additional Anopheles populations [11,55]. As this resistance mechanism may confer cross resistance to a wide range of contact insecticides, it is important that insecticide screening pipelines incorporate strains with thickened cuticles. However, our own observations indicate that this mechanism may be less stable in laboratory colonies than other resistance mechanisms, perhaps indicative of a high fitness cost which is balanced by other phenotypic advantages, such as ability to withstand desiccation [56], or mating advantage [10].

Further putative resistance mechanisms are indicated by examination of the RNAseq but have not been functionally validated. For example, two odorant binding proteins (AGAP000278 and AGAP012867) are up-regulated in all of the pyrethroid resistant populations from Burkina Faso. The chemosensory protein SAP2, expressed in mosquito legs and antennae, has already been shown to play a key role in pyrethroid resistance in *An. gambiae s.l* from Burkina Faso but [12], whilst OBPs have been associated with resistance in other studies [57,58], a direct role for this family in pyrethroid resistance remains to be demonstrated. Other gene families putatively involved in insecticide binding (and maybe sequestration) were elevated in multiple Burkina populations, most notably the hexamerins, found in the mosquito haemeolymph where they act as storage and transport proteins, which are highly enriched in the *An. arabiensis* resistant strain. The absence of DNA markers for these putative resistance mechanisms makes it difficult to evaluate their individual contributions to the phenotype but temporary loss of function via RNAi has been successfully used in the past to demonstrate a link between individual genes within putative insecticide binding protein families and resistance [40]. In vitro studies on recombinant proteins are also needed, both to confirm their role in pyrethroid binding, but importantly also to assess the ability to bind other insecticide classes.

## 5. Conclusions

This study demonstrates that different species within a species complex, collected from the same geographical area (including two originating from the same larval collections) and hence presumably under similar selection pressures, can evolve multiple, different resistance mechanisms. This may be indicative of the exceptionally strong selection pressure exerted on Anopheles mosquitoes in this major agricultural region in Burkina Faso but it presents a major challenge for existing and new insecticide based control tools. As the strains have been maintained under selection pressure in the laboratory, the fitness costs of alternative mechanisms, and hence their stability under natural settings, are unknown but nevertheless the strains represent a valuable biological resource for the screening of new insecticides for potential resistance liabilities. From an evolutionary perspective, genomic sequencing of these strains, coupled with further sampling of sympatric members of the species complex in the region, provides an opportunity to investigate the role of introgression versus de novo mutation, in the evolution of resistance, and in assessing the response to the introduction of ITNs with new classes of insecticides.

## Figures and Tables

**Figure 1 insects-13-00247-f001:**
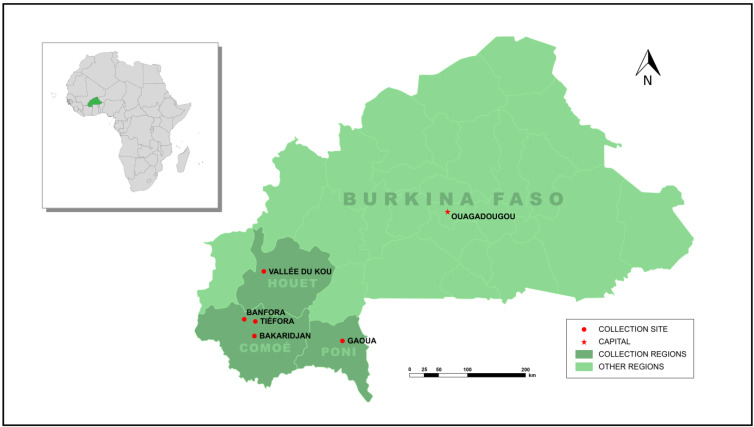
Map of Burkina Faso showing mosquito collection sites.

**Figure 2 insects-13-00247-f002:**
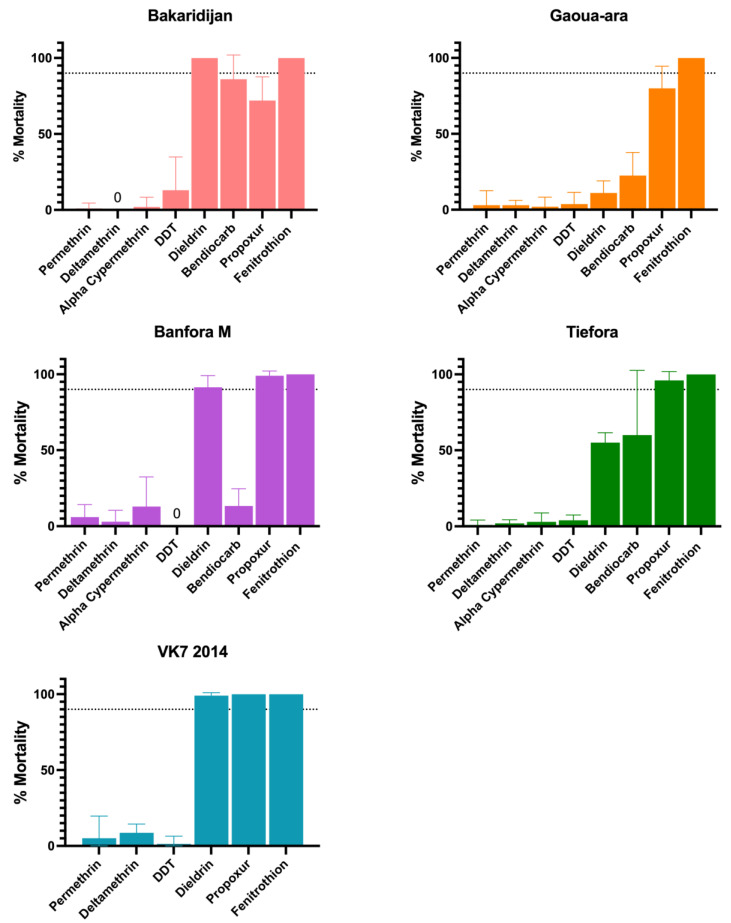
Mosquito mortality following exposure to insecticide papers in discriminating dose assays Mortality rates (%) 24 h after exposure for 5 strains of Anopheles mosquito (results shown from assays performed in 2019). Minimum sample size n = 80. Error bars represent 95% confidence intervals. Dotted line represents the WHO 90 % mortality resistance threshold.

**Figure 3 insects-13-00247-f003:**
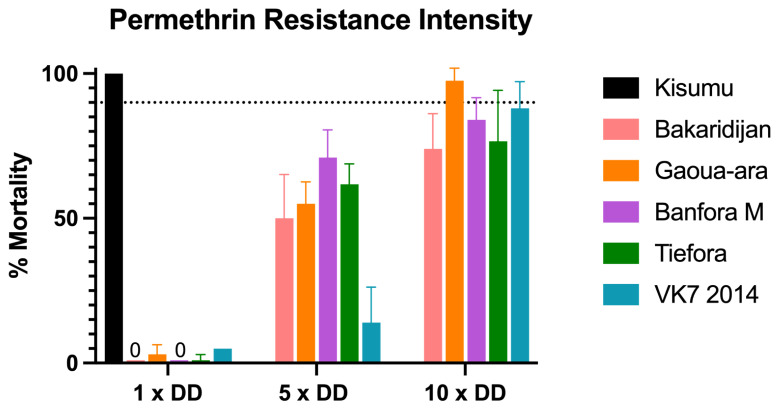
Mosquito mortality following exposure to permethrin papers in WHO resistance intensity assays. Mortality rates (%) 24 h after exposure for 5 strains of Anopheles mosquito. Minimum sample size n = 80. Error bars represent 95% confidence intervals. Dotted line represents the WHO 90% mortality resistance threshold. DD: Diagnostic dose.

**Figure 4 insects-13-00247-f004:**
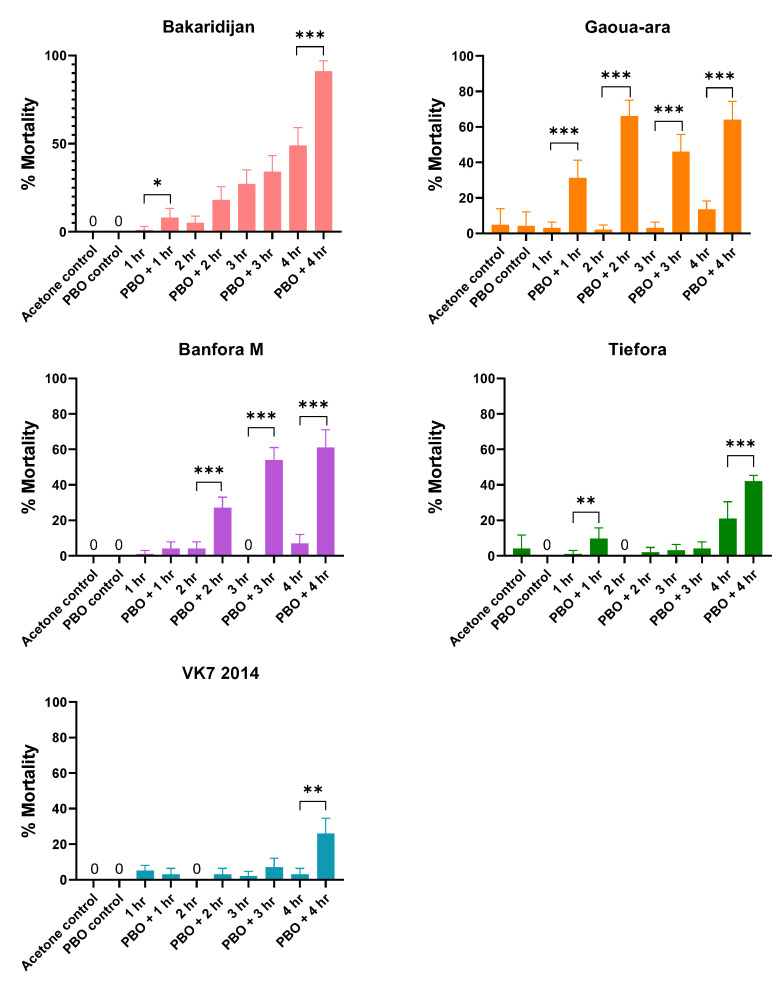
Mortality following exposure to permethrin with or without the synergist PBO. Mortality rates % 24 h after exposure. Minimum sample size n = 80. Error bars represent 95% confidence intervals. Statistical differences between permethrin only and PBO + permethrin for each paired combination indicated as * *p* <0.05, ** *p* < 0.001, *** *p* < 0.001 Fisher’s Exact test.

**Figure 5 insects-13-00247-f005:**
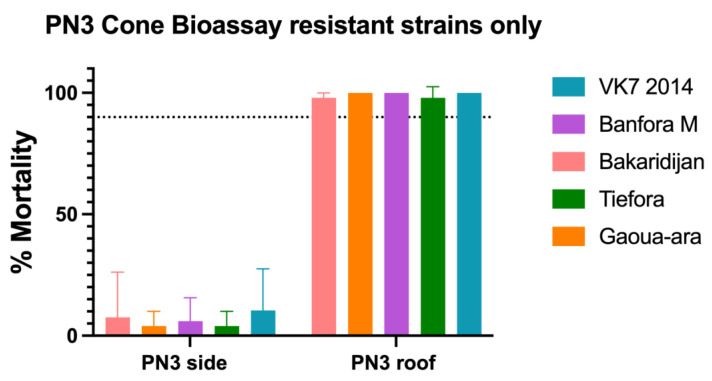
Mortality following exposure to Permanent 3.0 LLINs (PN3) in cone bioassays. Mortality rates % 24 h after exposure. Minimum sample size n = 50. Error bars represent 95% confidence intervals.

**Figure 6 insects-13-00247-f006:**
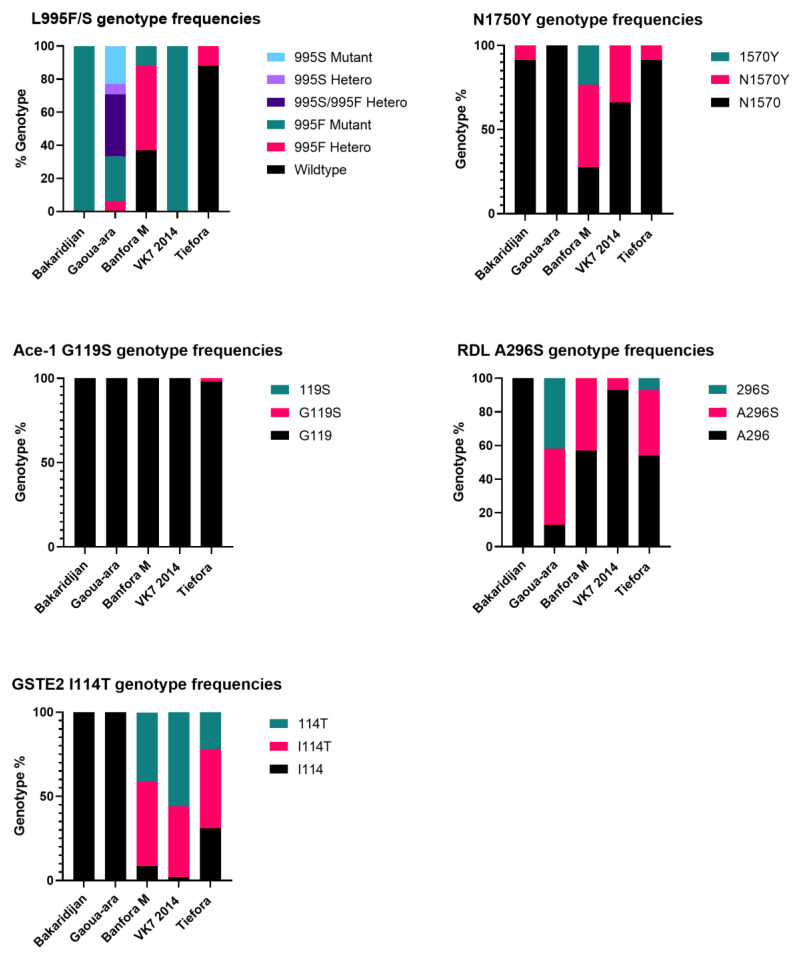
Frequency of point mutations associated with resistance. Data reported from samples genotyped in 2019. 995L, 1575N, 119G, 269A and 114I indicate the wildtype genotype (black bars); 995F, 995S, 1570Y, 119S, 296S and 114T indicate resistant genotype (green or purple bars). Heterozygote genotypes are shown with pink bars.

**Figure 7 insects-13-00247-f007:**
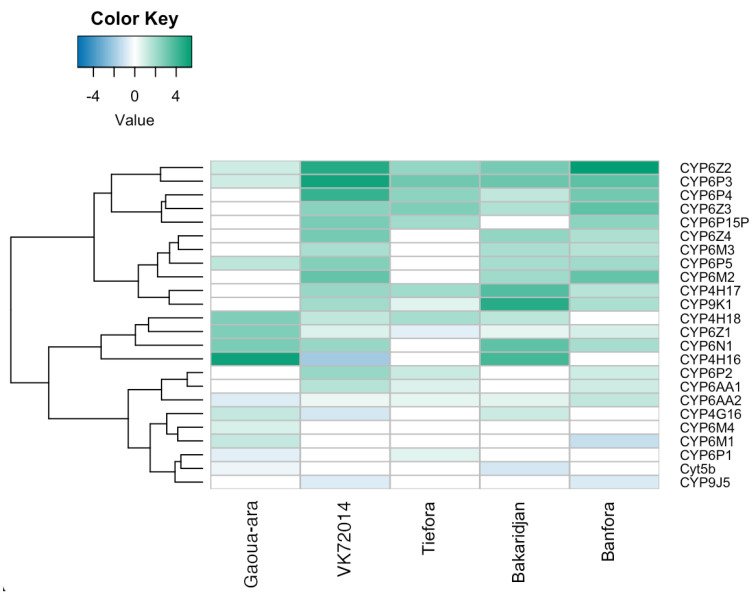
Heatmap showing cytochrome P450 genes that are significantly differentially expressed between the pyrethroid resistant and susceptible strains.

**Figure 8 insects-13-00247-f008:**
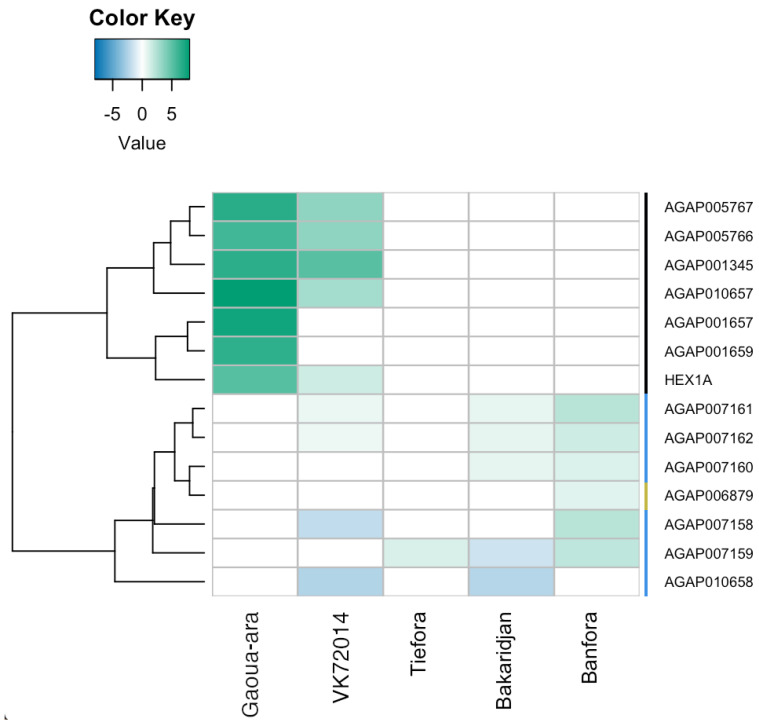
Heatmap showing differential expression of genes in families putatively associated with insecticide sequestration between the pyrethroid resistant and susceptible strains.

**Table 1 insects-13-00247-t001:** Summary of the *Anopheles gambiae* s.l. mosquito strains used in the study.

Strain	Species	Origin	Source	Year Colony Established
**Kisumu** **(susceptible strain)**	*An. gambiae* (s.s.)	Kenya	MR4	1975
**N’Gousso (susceptible strain)**	*An. coluzzii*	Cameroon	CRID	2006
**Moz** **(susceptible strain)**	*An. arabiensis*	Chokwe, Southern Mozambique (24° 33′ 37″ S, 33° 1′ 20″ E)	Established in LSTM from field collections performed by JCM with assistance from National Institute of Health, Mozambique [23]	2009
**VK7 2014**	*An. coluzzii*	Houet Province, Burkina Faso Valley de Kou 7 (11°24′29″ N, 4°24′37″ W)	Established from larval collections performed by LSTM (JCM) and CNRFP (KHT)	2015
**Banfora M**	*An. coluzzii*	Comoé Province Burkina Faso Banfora district (Tiefora, Pont Maurice, Sikane and Djomale (10° 38′ 0″ N, 4° 33′ 0″ W) and Bakaridjan (10°24′26.34″ N, 4°33′44.78″ W) villages)	Established from larval collections performed by LSTM (JCM) and CNRFP (KHT)	2015
**Bakaridjan**	*An. gambiae* (s.s)	Comoé Province Burkina Faso Banfora district (Tiefora, Pont Maurice, Sikane and Djomale (10° 38′ 0″ N, 4° 33′ 0″ W) and Bakaridjan (10°24′26.34″ N, 4°33′44.78″ W) villages)	Established from larval collections performed by LSTM (JCM) and CNRFP (KHT)	2015
**Tiefora**	*An. coluzzii*	Comoé Province, Burkina Faso Banfora district (10° 37.447’ N, 4° 33.201’ W)	Established from larval collections performed by LSTM (JCM) and CNRFP (KHT)	2018
**Gaoua-ara**	*An. arabiensis*	Poni Province Burkina Faso Gaoua district (10.3231° N, 3.1679° W)	Established from larval collections performed by IRSS (ASH)	2018

**Table 2 insects-13-00247-t002:** Summary of correlation between results of detoxification multiplex qPCR and RNAseq data.

	*An. coluzzii*			*An. gambaie*	*An. arabiensis*
	VK72014	Banfora	Tiefora	Bakaridjan	Gaoua-ara
**CYP4G16**					
**CYP6M2**					
**CYP6P1**					
**CYP6P3**					
**CYP6P4**					
**CYP6Z1**					
**CYP9K1**					
**GSTE2**					
	Genes up-regulated in both qPCR and RNAseq data set				
	Genes up-regulated in qPCR data set only				
	Genes up-regulated in RNAseq data set only				

## Data Availability

All RNAseq data is freely available on SRA under accessions PRJNA780362 and PRJNA750256. Custom code for enrichment analysis is available on GitHub at https://github.com/VictoriaIngham/BurkinsStrains.

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
