# Peer review of "Sympatric Populations of the Anopheles gambiae Complex in Southwest Burkina Faso Evolve Multiple Diverse Resistance Mechanisms in Response to Intense Selection Pressure with Pyrethroids"

_insects, 2022, doi:10.3390/insects13030247_

Round 1

Reviewer 1 Report

The manuscript from Williams et al. provides a very comprehensive description of the genetic background underlying insecticide resistance in mosquito populations colonized from southern Burkina faso and maintain in the insectary of Liverpool School of Tropical Medicine. The rationale of this work is well stated: because Anopheles populations from Sub-Saharan Africa have become highly resistant to pyrethroids, it has become urgent to develop and evaluate new chemicals for use in malaria vector control. Biological resources such as the pyrethroid-resistant strains established from different vector species will be extremely helpful to conduct screening. The paper is well written and the methodology is comprehensive, ranging from laboratory bioassays to detailed genome-wide analysis with RNAseq.

The only very minor concern I found in the fact that resistance is being maintained through selection in laboratory conditions. The mechanisms may differ between naturally occurring resistance and resistance selected in laboratory. Perhaps it would be important to conduct regular crosses after a certain number of generations between the lab strains and field populations from the original location.

Also the strains are likely susceptible to the new insecticides recently approved for mosquito control by WHO, but the authors didn’t show any susceptibility test to those new active ingredients.

Author Response

Response

We thank the reviewer for their positive comments on the manuscript.  They raise an interesting point about re-introduction of field material.  We recognise the advantages of this approach but, as one of the rationales for establishing these colonies was to provide a set of stable, fully characterised, populations, we feel that the reintroduction of field material might disrupt this goal.

We have tested all strains against chlorfenapyr and found them to be susceptible; a note has been added to the discussion.  Future work will characterise their susceptibility to other new active ingredients.  (see lines 418-423)

Reviewer 2 Report

Overview:

            The authors of this study used a comprehensive set of assays to test known and potentially previously undescribed insecticide resistance mechanisms in lab-established strains of the An. gambiae complex from Burkina Faso. Existing and relatively newly established mosquito strains were examined for resistance to multiple insecticides including multiple classes. Resistance to several insecticides was widespread, particularly pyrethroids. Exposure to piperonyl butoxide was able to restore the effectiveness of permethrin in several strains, and strain-specific differences with regard to the length of time of PBO exposure were noted. Insecticide-treated bed nets were also tested, but only the PBO+permethrin-treated roof of the nets induced high mortality in the resistant strains (in contrast to the permethrin only treated sides). Further screening of the strains was performed on known resistance alleles. In addition, RNAseq was employed to determine possible additional resistance mechanisms in resistant strains. P450s were examined in particular. Overall, this work was straightforward and well written and provided a broad look at insecticide resistance in established lab colonies of the An. gambiae complex in Burkina Faso.

I have only a few comments:

The caption for Supplementary figure 1 should define "Delta".

Line 336: "99FS" should be "995S"?

Lines 333 & 337: References should be provided for the "typically seen in resistant colonies" and "previously linked to insecticide resistance" statements.

Line 377: "highly" upregulated for CYP4G16 seems to be subjective dependent upon how the heatmap is constructed. In supplementary figure S5 this appears to be somewhat true for the Gouara and Bakaridjan strains, although the same transcript in Figure 7 is less convincing. Could the quantitative data be provided in a supplementary file?

Line 396: In addition to CYP6P3, CYP6Z1 also appears to have mixed agreement between qPCR and RNAseq, according to Table 3.

Line 510-511: "individual" should be deleted after "evaluate".

Author Response

The caption for Supplementary figure 1 should define "Delta".

Detail added

 Line 336: "99FS" should be "995S"?

 Corrected

Lines 333 & 337: References should be provided for the "typically seen in resistant colonies" and "previously linked to insecticide resistance" statements.

Done

Line 377: "highly" upregulated for CYP4G16 seems to be subjective dependent upon how the heatmap is constructed. In supplementary figure S5 this appears to be somewhat true for the Gouara and Bakaridjan strains, although the same transcript in Figure 7 is less convincing. Could the quantitative data be provided in a supplementary file?

We have added in the FC for the RNAseq data  for 4G16 and included the qPCR data as a new supplementary table (done qPCR data table)

 Line 396: In addition to CYP6P3, CYP6Z1 also appears to have mixed agreement between qPCR and RNAseq, according to Table 3.

Corrected

Line 510-511: "individual" should be deleted after "evaluate".

Corrected

Reviewer 3 Report

In the manuscript, by Williams and colleagues (Sympatric populations of the Anopheles gambiae complex in southwest Burkina Faso evolve multiple diverse resistance mechanisms in response to intense selection pressure with pyrethroids), the authors reported results from exhaustive bioassays experiments using newly established colonies of mosquitoes from the Anopheles gambiae complex to evaluate their pyrethroids resistance profile, phenotypically and genetically.

Some points to take into consideration:

It might be just a writing style issue; however, although English is not a problem here since it is grammatically perfect, I consider that the authors may need to invest more effort to explain the methods section a bit better. It was hard to follow/ read. Consider avoiding long sentences and semicolons and use final points instead. I think this might improve the reading quality of this section. For example:

Line 108 – 109: reads: Females were killed by freezing after oviposition 108 and the dried female, and egg papers were transported to the Liverpool School of Tropical 109 Medicine.

I think authors may consider it as: Females were killed by freezing after oviposition 108. Dried females and egg papers were transported to the Liverpool School of Tropical 109 Medicine.

Other points:

Title: Anopheles gambiae must be as full name.

Line 73: Italic “Anopheles gambiae s.l.” (s.l. should not be italicized)

 Line 80: Italic “An. gambiae s.l.

Line 82: Italic “An. gambiae s.l.” No need for s.l. when referring to the complex.

Line 83: Italic “An. gambiae s.s., An. colizzii and An. arabiensis”. Since it is the first time coluzzii and arabiensis appear in the text, please consider writing the full genus name.

Line 91: Italic “An. gambiae s.l.”

Please, italicize all scientific names and pay attention to the rules for the international nomenclature code of zoology. Be consistent with the names along with the whole text.

As I pointed out previously, it might be just a writing style issue but, saying “previously…” followed by the citation (number in this case) does not read well. Please consider using previously described by… or any other form.

Lines 91 – 92: Only these three members of the complex are found in Burkina Faso? Please provide some references here.

Lines 89 – 98: The authors give some short explanations about the methodology and results; however, it does not fit within the introduction. Very long and confusing paragraph. Please add Burkina Faso after the collection site names. I know it might sound redundant, then please consider reformulating.

Line 108: If “as described previously,” please add by who and then the citation.

Table 1: Hard to read. Head: does not have column names. Text overlap with text from adjacent columns. Coordinate format: replace semicolon for coma. Consider adding which strains are susceptible (to what) and which ones are resistances, as well as those under test.

Map aesthetics: North arrow missing. Scale: keep it up to 200km with three or four divisions. Make the names of the areas a bit clearer. The name of the county needs to be smaller. Please, keep the same font.

Line 126: consider, “Insectaries were maintained under standard conditions….”

Methods: Selection and resistance profiling:

Line 134: On what basis did the authors decide to use the insecticide paper up to 6 times? Was the insecticide at the same concentration? How were the papers handled to prevent insecticide degradation?

Lines 135 – 137: Not precise why exposure time varied between strains. Is there a previous report? Please provide more explanation about this method.

Overall: Although the authors cite articles describing the methodology, it is tough to picture what they have done. Please describe a bit clearer this section. Especially the eight insecticides used.

Methods: Conde bioassays:

Just a question: what was the idea to test a piece of the roof and a piece from the side? Why do you cut a piece or the net from the roof and side instead of just performing the cone bioassay in situ under normal natural conditions?

Methods: RNAseq transcriptomic analysis

Line 196: please cite reference genome

Line 198: read alignment?

Line 201: Consider final point.

Consider italicizing the name of the programs and functions used for analysis.

Results:

Were the susceptible strains (I supposed, used as controls) exposed to the same bioassays as the other strains? It would be nice to see results.

Author Response

Line 108 – 109: reads: Females were killed by freezing after oviposition 108 and the dried female, and egg papers were transported to the Liverpool School of Tropical 109 Medicine.

I think authors may consider it as: Females were killed by freezing after oviposition 108. Dried females and egg papers were transported to the Liverpool School of Tropical 109 Medicine.

We have corrected this instance and removed a few other semicolons throughout

Other points:

Title: Anopheles gambiae must be as full name.

Unclear what is meant here as this is written in full in the title

Line 73: Italic “Anopheles gambiae s.l.” (s.l. should not be italicized)

Corrected

 Line 80: Italic “An. gambiae s.l.

Corrected

Line 82: Italic “An. gambiae s.l.” No need for s.l. when referring to the complex.

Corrected

Line 83: Italic “An. gambiae s.s., An. colizzii and An. arabiensis”. Since it is the first time coluzzii and arabiensis appear in the text, please consider writing the full genus name

Corrected

Line 91: Italic “An. gambiae s.l.”

Corrected

Please, italicize all scientific names and pay attention to the rules for the international nomenclature code of zoology. Be consistent with the names along with the whole text.

Checked throughout

As I pointed out previously, it might be just a writing style issue but, saying “previously…” followed by the citation (number in this case) does not read well. Please consider using previously described by… or any other form.

We have not made this change as we feel it is clear as written

Lines 91 – 92: Only these three members of the complex are found in Burkina Faso? Please provide some references here.

New reference (Namountougou et al) added

Lines 89 – 98: The authors give some short explanations about the methodology and results; however, it does not fit within the introduction. Very long and confusing paragraph. Please add Burkina Faso after the collection site names. I know it might sound redundant, then please consider reformulating.

We have added Burkina Faso as requested.  We feel this paragraph summarises the key points of the manuscript and would prefer to keep as is if the editors are in agreement.

Line 108: If “as described previously,” please add by who and then the citation.

Added

Table 1: Hard to read. Head: does not have column names. Text overlap with text from adjacent columns.

This appears to have been a formatting error introduced after uploading. Column names did appear originally, and text didn’t overlap.  We have attempted to re-format to avoid further issues.

Coordinate format: replace semicolon for coma.

Corrected

Consider adding which strains are susceptible (to what) and which ones are resistances, as well as those under test.

We have indicated which of the strains in the table are susceptible.

Map aesthetics: North arrow missing. Scale: keep it up to 200km with three or four divisions. Make the names of the areas a bit clearer. The name of the county needs to be smaller. Please, keep the same font.

Updated with the above suggestions

Line 126: consider, “Insectaries were maintained under standard conditions….”

Corrected

Methods: Selection and resistance profiling:

Line 134: On what basis did the authors decide to use the insecticide paper up to 6 times? Was the insecticide at the same concentration? How were the papers handled to prevent insecticide degradation?

This is the standard procedure used by the World Health Organisation, the relevant reference has been added

Lines 135 – 137: Not precise why exposure time varied between strains. Is there a previous report? Please provide more explanation about this method.

As already explained in the text, exposure times varied between strains to ensure at least 50% survival

Overall: Although the authors cite articles describing the methodology, it is tough to picture what they have done. Please describe a bit clearer this section. Especially the eight insecticides used.

Details of the 8 insecticides used has been added into the text.

Methods: Conde bioassays:

Just a question: what was the idea to test a piece of the roof and a piece from the side? Why do you cut a piece or the net from the roof and side instead of just performing the cone bioassay in situ under normal natural conditions?

The standard WHO cone assay was adopted for this work; the reference is included.  

Methods: RNAseq transcriptomic analysis

Line 196: please cite reference genome

We are not clear what additional information is needed here as the text already states aligned to the latest Anopheles gambiae s.l. genome assembly PEST4

Line 198: read alignment?

Again, we are not sure what is needed here as the text already states Over 70% read assignment was seen for each replicate of each population

Line 201: Consider final point.

Consider italicizing the name of the programs and functions used for analysis.

We can make this change if it is editorial practice for the journal

Results:

Were the susceptible strains (I supposed, used as controls) exposed to the same bioassays as the other strains? It would be nice to see results.

Details of results from susceptible strains have been added